# Investigation into the Rationale of Migration Intention Due to Air Pollution Integrating the *Homo Oeconomicus* Traits

**Quan-Hoang Vuong** [1] , **Tam-Tri Le** [1,2] , **Viet-Phuong La** [1,*] , **Thu-Trang Vuong** [2]
and **Minh-Hoang Nguyen** [1,*]

[1] Centre for Interdisciplinary Social Research, Phenikaa University, Ha Dong District, Hanoi 100803, Vietnam; hoang.vuongquan@phenikaa-uni.edu.vn (Q.-H.V.); tri.letam@phenikaa-uni.edu.vn (T.-T.L.)

[2] A.I. for Social Data Lab (AISDL), Vuong & Associates, Dong Da, Hanoi 100000, Vietnam

[*] Correspondence: phuong.laviet@phenikaa-uni.edu.vn (V.-P.L.); hoang.nguyenminh@phenikaa-uni.edu.vn (M.-H.N.)

**Abstract:** Air pollution is a considerable environmental stressor for urban residents in developing countries. Perceived health risks of air pollution might induce migration intention among inhabitants. The current study employed the Bayesian Mindsponge Framework (BMF) to investigate the rationale behind the domestic and international migration intentions among 475 inhabitants in Hanoi, Vietnam—one of the most polluted capital cities worldwide. We found that people perceiving more negative impacts of air pollution in their daily life are more likely to have migration intentions. The effect of perceived air pollution impact on international migration intention is stronger than that of domestic migration. Acknowledging a family member's air pollution-induced sickness moderated the association between perceived air pollution impact and domestic migration intention, while the personal experience of air pollution-induced sickness did not. In contrast, the moderation effect of personal experience of sickness became significant in the international migration circumstance, but the effect of information about a family member's sickness was negligible. The findings suggest that urban inhabitants' consideration of air pollution-averting strategies reflects some characteristics of *Homo Oeconomicus*. Although an individual's socioeconomic decision may seem insignificant on a collective scale, through environmental stressors as catalysts, such decisions might result in considerable social tendencies (e.g., internal migration and emigration).

**Keywords:** air pollution; migration intention; BMF analytics; *Homo Oeconomicus*; Vietnam

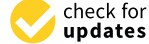



## 1. Introduction

Nobody wants to live in a harmful environment if they can choose a better alternative option. This is a natural tendency observed in any evolutionary level of biological organisms. Negative chemotaxis (the migration of microorganisms in a direction corresponding to a chemical gradient) is observed in certain bacterium species as they move away from harmful substances [1,2]. More complex organisms have more complex interactions with their surroundings; for example, bees are particularly sensitive to environmental stressors which damage their nervous system and lead to cognitive dysfunctions [3]. Many species migrate to places with better conditions in the animal kingdom, especially when their current habitat is unsuitable for survival or reproduction (e.g., due to seasonal change). Migration behavior is phenotypically plastic and depends on environmental conditions such as temperature [4]. However, displacement is unfortunately not always due to natural reasons.

Pollution from human activities is one of the main reasons for habitat loss [5], making many areas no longer suitable for sustaining certain species. Humans are also biological organisms that naturally aim for optimal living conditions while avoiding harmful factors if possible. However, arguably, unlike other species, humans use a much more complex

rationale when deciding their responses to perceived risks [6] because a person's behaviors are determined by intentions based on beliefs and perceptions [7]. When people consider whether to migrate due to pollution, what would be the reasons behind their decision?

People can develop averting behaviors as a defensive response to the environmental stress caused by pollution. Such coping responses in a polluted urban environment vary as people may try to adapt by limiting outdoor exposure [8,9] and making changes to their everyday activities, such as skipping school [10] and reducing outdoor cycling [11] and public park visits [12]. People may decide to migrate on a larger scale due to environmental stress [13]. Areas with high levels of air pollution were found to have decreased immigration rates [14] and immigrants' willingness to stay [15]. Within a country, people tend to migrate away from regions with relatively worse air quality [16–18]. People who are dissatisfied with their city's air quality are more likely to have internal migration intentions [19].

Environmental stress can also induce international migration, and the cost–benefit approach for migrating decisions is important if we consider migrants' ability to choose among available alternatives [20]. A higher level of migration-related information-seeking activities was found to follow a higher measured level of air pollution from the day before [21], which shows how the emigration interest is influenced by information inputs regarding the pollution status of one's living environment.

Theories on migration also point to the important role of the cost–benefit assessment when people decide to migrate. In the late 19th century, Ravenstein proposed the "Laws of Migration", which stated the negative association between the distance and volume of migration, among other generalizations [22]. It was followed by decades of research and theoretical development, such as Stouffer's theory that migration volume is proportional to the opportunities (advantages) at the destination [23]. In 1966, Everett S. Lee proposed a more comprehensive theory of migration [24]. Lee suggests that migration decision is based on the net value of positive, neutral, and negative factors involving the origin, destination, intervention, and personal contexts. These factors are assessed differently from person to person, depending on individual perceptions. Whether to favor the act of migrating or not is the result of multifaceted cost–benefit judgments.

Intuitively, people are against living in a place with low air quality. Air pollution does not only lead to subjective concerns and annoyance [25], but there are also various serious health risks, both physical [26] and mental [27,28]. The productivity of physical laborers [29] and white-collar workers [30] can decline as a result of air pollution. Exposure to air pollution has been linked to general psychological distress [27], depressive disorder [28,31], and even suicide [32–34]. It also negatively affects cognitive performance in adults [35,36], particularly in the elderly, and is a risk factor for dementia [37,38]. Even worse, in the current pandemic, air pollution was also found to be a factor that increases the likelihood of COVID-19 death [39]. It is also worth mentioning that even at non-toxic exposure levels, a person's perception of air pollution and health risk can lead to discomfort and health symptoms [25]. Empirically, risk perception acts as a mediator for migration decisions driven by environmental hazards [40]. For these reasons, perceived health consequences induced by air pollution can play important roles in people's cost–benefit judgements toward migration.

Although environmental stress influences the intention to migrate, it is not the only factor in the complex consideration for such action. Regarding the theoretical rationale leading to migration intention, humans have many other motives besides health concerns, and finance is a major one. The original approach of *Homo Oeconomicus* ("Economic Man") proposed by John Stuart Mill suggests that humans rationally aim for optimal personal economic benefits based on self-interested desires [41]. This view of a strictly rational decision-making model assumes that an individual has specific predetermined personal goals and will try to attain them at the lowest cost possible. The *Homo Oeconomicus* approach has received a lot of criticism, mainly for the implied dominant selfishness in its original conceptualization [42,43], and studies have provided evidence for adjustment

and expansion of the original theory [44]. While the view has its flaws, it can help hint at a more integral cost–benefit model of subjective reasoning for human behavior. When deciding to migrate due to air pollution, the *Homo Oeconomicus* assesses that such action can have a positive net benefit after considering both health and financial aspects.

Migration is costly in one way or another, and not just for humans. For example, migration can hold risks for insects across many aspects, such as energy consumption, predation, or reproduction [45]. When choosing places with suitable opportunities for the same purpose, people perceive migrating over a longer distance to be more costly—physically and financially [43], as well as in terms of diminishing information [46]. Those searching for another city with better air quality are more likely to choose a nearby city rather than a faraway one as their destination [19]. When migrating to a new environment, the cost of distance is physical and should consider the aspect of "cultural distance", or in other words, the complex effect of culture shock [47]. Nevertheless, to our knowledge, no study has employed the *Homo Oeconomicus* approach to reason the migration intention due to environmental stress.

Vietnam is a developing nation with one of the most hazardous levels of air quality in the Asia-Pacific region [48]. Hanoi city—where we collected survey data for the present study—is considered one of the most polluted capital cities in the world [49], with air pollutants mainly from traffic, industrial emissions, and construction sites [50] that pose a considerable risk of respiratory and cardiovascular problems for the citizens [51]. An assessment of the health risk caused by mobility in Hanoi reveals that $PM_{10}$ emitted by traffic can cause 3200 fatalities annually [52]. Using data on daily admissions from the Vietnam National Hospital of Pediatrics and daily records of air pollution, Luong et al. [44] discover that increasing levels of particulate matter (e.g., $PM_{10}$, $PM_{2.5}$, or $PM_1$) are associated with respiratory admissions of young children in Hanoi. Given the severe air pollution in Vietnam, specifically Hanoi, some studies have been conducted on air-pollution-induced migration [19,53]. However, those studies do not examine how domestic and international migration intentions can be affected by the perceived health consequences of the self and other family members.

For this reason, we examine how the likelihood of having domestic and international migration intentions can be driven by the perceptions of the individual's and family members' health consequences induced by air pollution. To fulfill this research objective, we employed the Bayesian Mindsponge Framework (BMF) analytics), which combines the mindsponge theory's reasoning strengths and the inference advantages of Bayesian analysis. The estimated results reveal some signs of *Homo Oeconomicus* in Vietnamese urban residents' psychological process of migration intention. Thus, the result elaboration through the lens of *Homo Oeconomicus* is presented in the Discussion.

## 2. Methodology

### 2.1. Theoretical Foundation and Model Construction

This subsection describes the mindsponge theory, on which the information-processing mechanism would be grounded for model construction [54,55]. The theory was developed from the mindsponge mechanism, which is an inclusive model of cognition-shifting processes that demonstrates how new information, deemed valuable, is absorbed from the external environment while information deemed valueless or with waning values is ejected from the individual's mind [56,57]. The original mindsponge mechanism is built by observing psychological and social phenomena, which is also true for many other theories and frameworks, including those developed by Abraham Maslow [58], Geert Hofstede [59], Inoue Nonaka [60], Henry Mintzberg [61], and Michael Porter [62], etc. Although this approach has proven useful for comprehending the complexity of human behavior and social systems, it still does not account for the fundamental elements that make a human a human: the cells and molecules and the processes that give life to both inorganic and organic compounds. Thus, the mindsponge mechanism was upgraded into mindsponge

theory using the newest evidence from brain and life sciences [55]. The theory has been used in many studies investigating psychological and behavioral issues [63–72].

The theory suggests that the human mind is an information collection-cum-processor. For a mind to function, it requires physical structures that serve as platforms for processing activity, such as human brains. The mindset is the collection of all accepted knowledge in the system, which takes memory to retain. Based on the content of the present mindset, the filtering mechanism determines what information enters and exits the mindset, which updates the mind and subsequent filtering system. This process is dynamically balanced and involves cost–benefit evaluation, which aims to increase the perceived benefit and reduce the perceived cost of the system. Specifically, if the information is perceived to be beneficial, it will be absorbed into the mindset. In contrast, the information will be dismissed from the mind when perceived as costly (in other words, if its net perceived benefit is negative). Therefore, ideation can only occur within the mind when the idea is deemed beneficial.

Following this logic of mindsponge theory, we assume that migration ideation due to air pollution is more likely to emerge in the mind when an individual feels the negative impacts of air pollution on their life. In other words, when an individual becomes more aware of air pollution's adverse impacts on their life, they tend to find alternatives to minimize the cost of air pollution, including migrating to other geographical locations. Thus, we postulate that the more an individual perceives the impacts of air pollution on their life, the more likely they will think of migrating to another city or country.

Perceived air pollution impacts can be myriad. Urban residents might attribute dust and smog, their uncomfortable feelings, shortness of breath, and respiratory diseases to the consequences of air pollution. Among these consequences, respiratory diseases are a severe impact of air pollution on an individual's health. Thus, we postulate that the association between perceived impacts of air pollution and migration intention can also be positively moderated by (1) the individual's attribution of their respiratory diseases to the consequence of air pollution and (2) knowing that a family member has been sick because of air pollution. This postulation can be explained in two ways.

First, prior experience or knowledge of respiratory disease attributed to air pollution can affect the individual's trust evaluation process. In particular, it can increase trust, or certainty, towards the information regarding the negative effects of air pollution (e.g., economic and health losses) and reduce the evaluation rigor towards information related to air pollution countermeasures, which increases the emergence probability of migration ideation. Second, when individuals suffer from respiratory disease, they might be absent from work, or their working productivity might decline. Therefore, experiencing a respiratory disease might lead to income loss because of decreasing productivity or "discontinued operations". Such income loss adds to the cost–benefit judgment of the individual and increases the cost of staying (or increases the benefit of moving). The experiences of one's own or a family member's air-pollution-induced health problems are treated as information pieces that hold different types of net costs (including financial costs, opportunity costs, etc.) in the subjective cost–benefit assessment—not only the direct cost of health risks.

For these reasons, we constructed the prediction model of urban residents' migration ideation as follows:

$$MoveCity \sim normal(\mu, \sigma) \tag{1}$$

$$\mu_i = \beta_0 + \beta_{ImpactDegree} * ImpactDegree_i + \beta_{PersonalHealth * ImpactDegree} * PersonalHealth_i * ImpactDegree_i \\ + \beta_{FamilyMemberHealth * ImpactDegree} * FamilyMemberHealth_i * ImpactDegree_i \tag{2}$$

$$\beta \sim normal(M, S) \tag{3}$$

The probability around $\mu$ is determined by the form of the normal distribution, whose width is specified by the standard deviation $\sigma$. $\mu_i$ indicates urban resident $i$'s probability of having the intention to move their family and work to a less polluted province due to air pollution concerns; $ImpactDegree_i$ indicates urban resident $i$'s perceived impacts of air

pollution on their life; $PersonalHealth_i$ indicates whether urban resident $i$ was sick due to air pollution; $FamilyMemberHealth_i$ indicates whether urban resident $i$'s family member was sick due to air pollution. Model 1 (displayed by Equations (1)–(3) has five parameters: the coefficients, $\beta_{ImpactDegree}$, $\beta_{PersonalHealth*ImpactDegree}$, and $\beta_{FamilyMemberHealth*ImpactDegree}$, the intercept, $\beta_0$, and the standard deviation of the "noise", $\sigma$. The coefficient of the variable $ImpactDegree_i$ is distributed as a normal distribution around the mean denoted $M$ and with the standard deviation denoted $S$. $\beta_{PersonalHealth*ImpactDegree}$ indicates the coefficient of the non-additive effect of $PersonalHealth_i$ and $ImpactDegree_i$ on $MoveCity$. If the coefficient $\beta_{PersonalHealth*ImpactDegree}$'s distribution is significant, and the association between the perceived impacts of air pollution and intention to move to a less polluted province is considered conditional on the person's experience of air-pollution-induced sickness. A similar interpretation is applied for $\beta_{FamilyMemberHealth*ImpactDegree}$. It should be noted that the distribution here is the distribution of parameter values but not the distribution of data. More details regarding the difference between these two distribution types are available in [73].

When deciding to move, the distance from the current location to the speculated destination can also influence the probability of migrating because of the transportation cost. Even though the adverse air pollution impacts might be migration incentives, these incentives might decrease if the moving cost is expensive, especially if the migration destination is abroad. Moving to a foreign country induces higher economic, psychological, and physical costs than domestic migration. Acculturation is a process in which an individual has to adjust and adapt to the new culture after migration. While undergoing such processes, acculturative stress inflicted by culture shocks might arise and negatively affect the individual's psychological and physical well-being. As these costs are higher when migrating abroad than when migrating domestically, we suspected that the effects of perceived air pollution impacts, prior experience, and acknowledgment of respiratory disease attributed to air pollution on the probability of moving to a foreign country would be lesser than the probability to move to another province. Thus, Model 2 (displayed by Equations (4)–(6)) was constructed:

$$MoveCountry \sim normal(\mu, \sigma) \tag{4}$$

$$\mu_i = \beta_0 + \beta_{ImpactDegree} * ImpactDegree_i + \beta_{PersonalHealth*ImpactDegree} * PersonalHealth_i * ImpactDegree_i \\ + \beta_{FamilyMemberHealth*ImpactDegree} * FamilyMemberHealth_i * ImpactDegree_i \tag{5}$$

$$\beta \sim normal(M, S) \tag{6}$$

The probability around $\mu$ is determined by the form of the normal distribution, whose width is specified by the standard deviation $\sigma$. $\mu_i$ indicates urban resident $i$'s probability of having the intention to move their family and work to a less polluted foreign country due to air pollution concerns.

### 2.2. Variable Selection

In this study, we used the data retrieved and combined from two open datasets about Hanoi inhabitants' perceptions of air pollution [74,75]. The two datasets were procured by combining two different survey collections in the central and suburban areas of the city. Both surveys were collected using stratified random sampling methods to ensure the representativeness of the dataset. There was a total of 475 respondents. A majority of the samples were in the age group of 19–40 (52.84%). Male respondents constitute 54.53% of the dataset, while female respondents constitute 45.26%. The educational level of most of the respondents was an undergraduate degree (55.79%); 4.42% of the respondents acquired master's or doctoral degrees; 38.95% of the respondents acquired an educational level lower than a bachelor's degree.

The survey collections were organized in November and December of 2019. Hanoi was chosen as the study site for the following three reasons: (1) Hanoi was ranked 7th

among the most polluted capital cities around the world [49]; (2) Hanoi is one of the fastest-growing cities in Vietnam; (3) Hanoi is the second largest and most populous city in Vietnam.

According to Khuc, Phu, and Luu [74], the survey collection comprises three steps. First, the collectors were recruited and well-paid to encourage them to perform well during the collection process. The researchers also held two four-hour seminars to help the collectors understand the project's goals and the questionnaire before the collection began. During the seminars, essential skills and techniques were instructed to collectors to obtain targeted information from respondents. The final version of the questionnaire was ensured to be error-free, straightforward, and easy to understand by conducting two pilot tests. Lastly, the collectors conducted face-to-face interviews with the respondents and maintained mutual interaction and communication to solve issues or questions arising during the survey collection.

Within the dataset, we purposely chose variables that could help explain the immigration intentions of Hanoi inhabitants due to air pollution. In total, we employed five variables for two models: two outcome variables and three predictor variables. The description of each variable is presented in Table 1.

**Table 1.** Variable description.

| Variable | Meaning | Type of Variable | Value |
|----------|---------|------------------|-------|
| *MoveCity* | Whether the respondent had the intention to move their family and work to a less polluted province due to air pollution concerns | Binary | No = 0 Yes = 1 |
| *MoveCountry* | Whether the respondent had the intention to move their family and work to a less polluted foreign country due to air pollution concerns | Binary | No = 0 Yes = 1 |
| *PersonalHealth* | Whether the respondent was sick because of air pollution | Binary | No = 0 Yes = 1 |
| *FamilyMemberHealth* | Whether any member of the respondent's family was sick because of air pollution | Binary | No = 0 Do not know = 0 Yes = 1 |
| *ImpactDegree* | The respondent's perceived impacts of air pollution on their life | Ordinal | From 1 (very impacted) to 4 (not impacted) |

The outcome variable in Model 1 is *MoveCity*, generated from the question "Are you planning to move your family and work in another province with less pollution?", with 430 "no" answers (90.53%) and 44 "yes" answers (9.26%). The outcome variable in Model 2 is *MoveCountry*, generated from the question "Are you planning to move your family to a less polluted foreign country?", with 439 "no" answers (92.42%) and 36 "yes" answers (7.58%). Regarding the variable *ImpactDegree*, the numbers of answers of values from 1 to 4 are 201 (42.32%), 232 (48.84%), 33 (6.95%), 8 (1.68%), respectively. It can be seen that most surveyed residents perceive rather high levels of impact from air pollution. Regarding whether respondents were sick because of air pollution, there were 234 "yes" answers (49.26%) and 241 "no" answers (50.47%). Regarding whether members of the respondents' families were sick because of air pollution, there were 238 "yes" answers (50.11%) and 233 "no" answers (49.05%). Note that a few missing data points would be omitted in the analysis.

### 2.3. Analysis and Validation

The Bayesian Mindsponge Framework (BMF) was employed as the methodological design for the current study [76,77]. BMF is an analytical framework that combines the advantages of the mindsponge theory and Bayesian inference to investigate humans' psychological and behavioral concepts or phenomena. In the framework, the mindsponge theory, with its ability to reflect the complexity and dynamics of a human mind, is used to construct theoretical models. At the same time, the Bayesian inference, with its high flexibility, helps fit those models for statistical analysis. The two parts are highly compatible and benefit each other in the process of conducting a study. In brief, their match has the following main points: (1) both mindsponge and Bayesian inference are built on subjectivity at philosophical and theoretical levels, which is fitting for social and especially psychological research; (2) they give researchers great flexibility in model construction and fitting; and (3) both can operate with an updating manner.

As the human psychological process is highly complex, we determined to construct parsimonious models to improve predictability. The Bayesian inference approach is good for estimating parsimonious models because it probabilistically treats all the properties, including the unknown parameters. Moreover, Bayesian analysis aided by the Markov Chain Monte Carlo (MCMC) algorithm offers the capability to estimate models with high complexity, such as those in the current study with non-linear relationships. Estimating the non-linear relationships makes the model more complex and requires a larger sample size for sound estimation [78]. A large number of iterative samples generated by the stochastic processes of Markov chains can help fit complex models effectively.

It is important to note that science is currently facing a reproducibility crisis that many studies across different fields, especially psychology [79] and social sciences in general [80], could not (or could not easily) be replicated due to technical issues involving the employed analytic approach. The wide sample-to-sample variability in the *p*-value is suggested to be the main reason for the crisis [81]. Thus, avoiding the use of *p*-value is another reason we employed Bayesian analysis because Bayesian analysis allows us to interpret results based on credible intervals.

Lastly, prior distribution incorporation is an advantage of Bayesian analysis. Even though we set priors as "uninformative" to avoid the subjective influences over the simulated outcomes, the prior function can still be capitalized to check the robustness of the simulated results by performing the "prior-tweaking" technique.

For validating the simulated posterior outcomes, we adopt a three-pronged validation strategy. Initially, we used Pareto-smoothed importance sampling leave-one-out cross-validation (PSIS-LOO) diagnostic plots to check the goodness-of-fit on every simulated model [82]. The model can be deemed a good fit with the data if the *k* values shown on the plot are all below 0.5. Next, we continued with the convergence check using diagnostic statistics and plots. The diagnostic statistics include the effective sample size (*n_eff*) and the Gelman–Rubin shrink factor (*Rhat*), while the diagnostic plots include the trace, Gelman–Rubin–Brook, and autocorrelation plots. Finally, the prior-tweaking technique was performed. Details of diagnostic statistics and plots are presented with explanation and interpretation in the Results section.

The **bayesvl** R package was used to conduct Bayesian analysis in this study [83,84]. For easy and transparent replication or cross-checking, the dataset, data description, and code snippets of Bayesian analysis were all deposited on The Open Science Framework (https://osf.io/us5tr/ (accessed on 3 July 2022)).

## 3. Results

### 3.1. Model 1: Intention of Immigration to Other Provinces

Model 1 examines the predictions of the perceived impact of air pollution and its interactions with prior negative experiences with air pollution towards immigration intention to a less polluted province. Model 1's logical connection is shown in Figure 1. The arrows show the directions of influence toward the outcome variable (here is *MoveCity*). Where

one line of direction involves two independent variables (two arrows before reaching the outcome variable), it indicates a non-linear influence from the interaction of an independent variable with another.

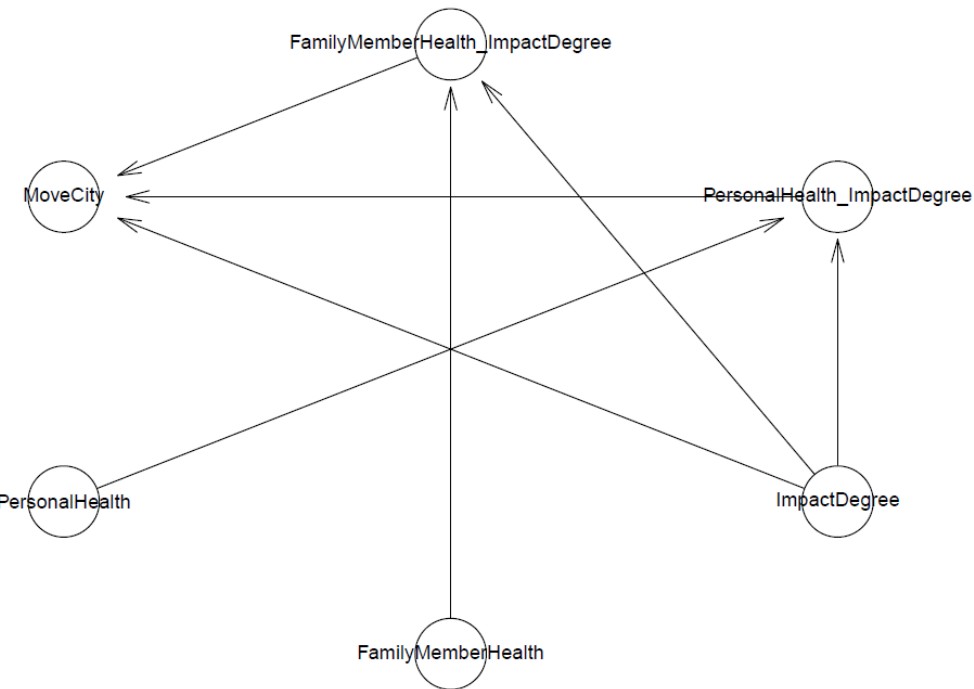

**Figure 1.** Model 1′s logical network.

The PSIS diagnostic plot shows that all *k* values are below 0.5, suggesting that Model 1 has a high goodness-of-fit with the data (see Figure 2).

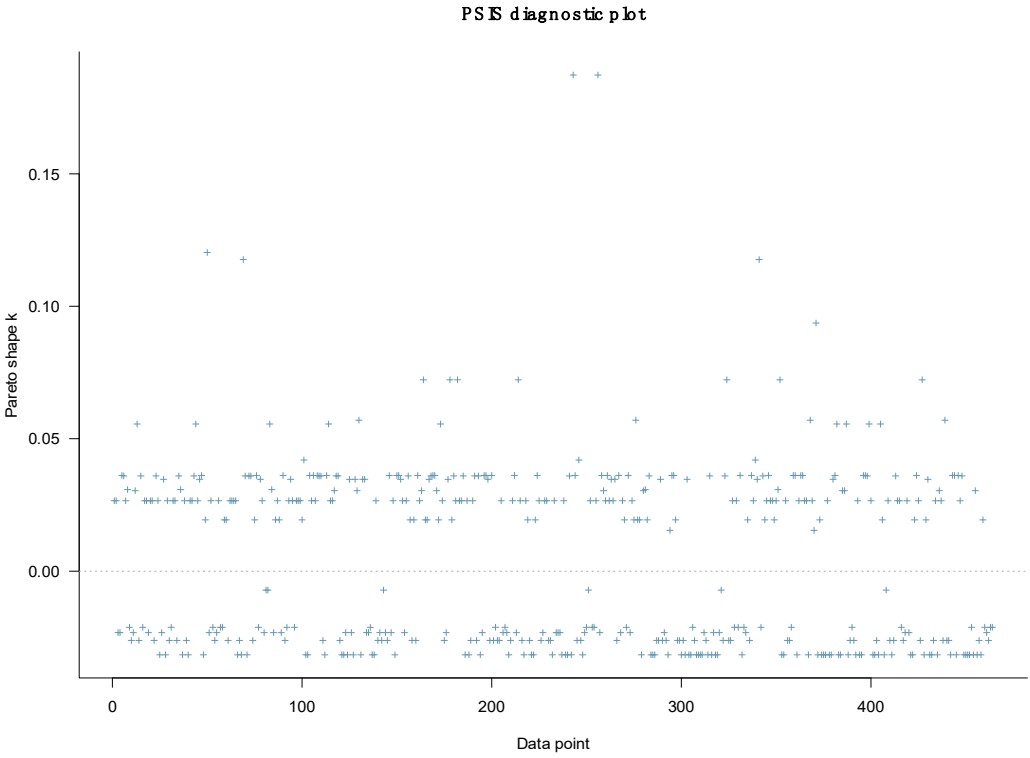

**Figure 2.** Model 1′s PSIS diagnostic plot.

The diagnostic statistics portray a good convergence of the model's Markov chains; the effective sample size (*n_eff*) statistics are larger than 1000, and Gelman shrink factor (*Rhat*) statistics are equal to 1 (see Table 2). The convergence is also confirmed by the trace plots, autocorrelation plots, and Gelman plots.

**Table 2.** Model 1's simulated posteriors.

| Parameters | Uninformative | | Belief | | Disbelief | | *n_eff* | *Rhat* |
|---|---|---|---|---|---|---|---|---|
| | Mean | SD | Mean | SD | Mean | SD | | |
| *Constant* | −1.90 | 0.49 | −2.00 | 0.48 | −1.89 | 0.48 | 4921 | 1 |
| *ImpactDegree* | −0.48 | 0.29 | −0.52 | 0.30 | −0.46 | 0.29 | 4933 | 1 |
| *PersonalHealth ∗ ImpactDegree* | 0.02 | 0.23 | 0.15 | 0.20 | 0.03 | 0.20 | 7545 | 1 |
| *FamilyMemberHealth ∗ ImpactDegree* | 0.35 | 0.26 | 0.41 | 0.20 | 0.30 | 0.20 | 7541 | 1 |

Note: SD = Standard deviation; The effective sample size (*n_eff*) and Gelman value (*Rhat*) of simulated results with different priors are almost similar, so only the *n_eff* and *Rhat* of simulated results using uninformative priors are presented.

Trace plots of all posterior parameters are presented in Figure 3. The *y*-axis of the trace plot represents the posterior values of each parameter, while the *x*-axis represents the iteration order of the simulation. The Markov chains are the colored lines in the middle of the trace plot. Markov chains can be considered good-mixing and stationary if they fluctuate around a central equilibrium. These two characteristics are a good signal of convergence.

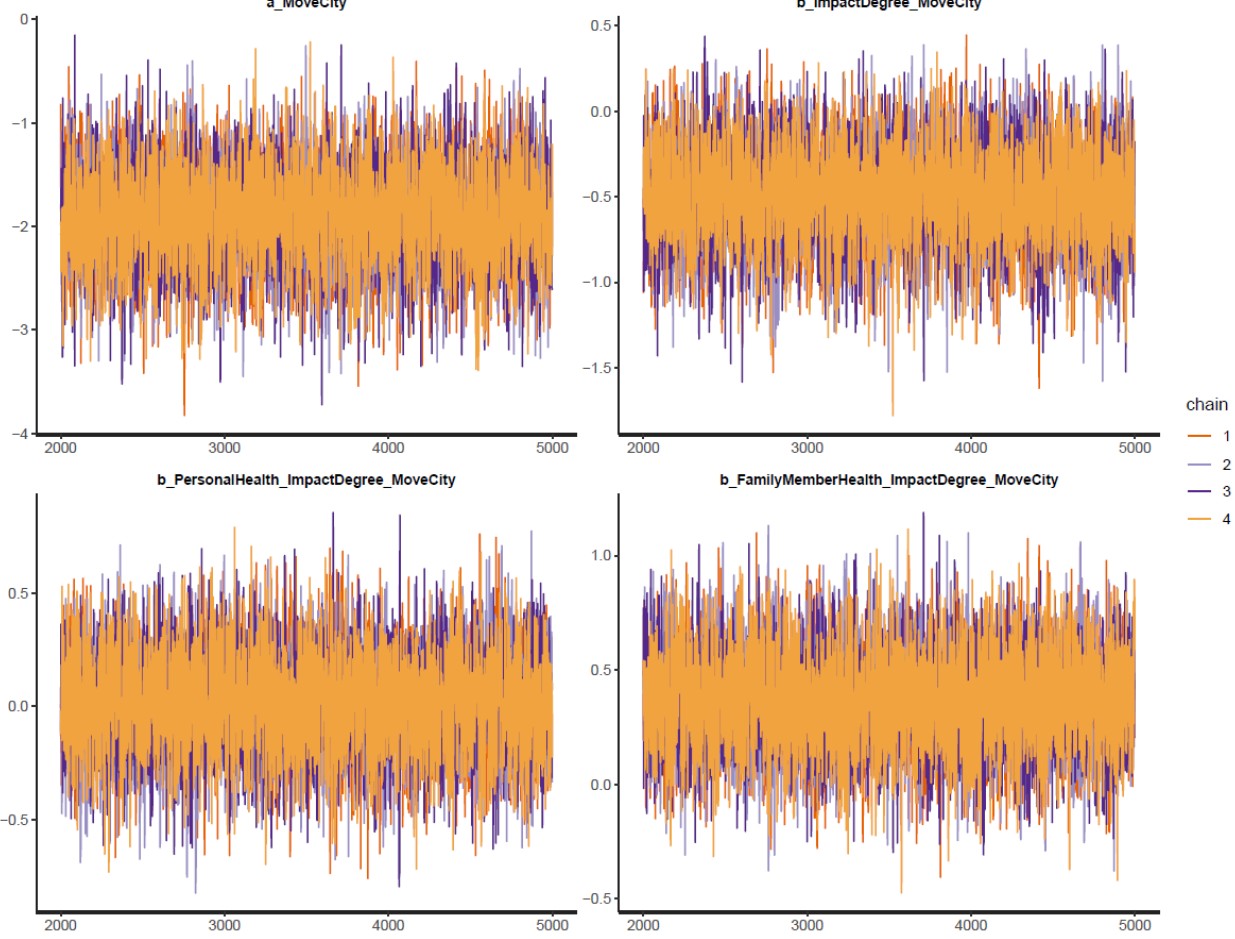

**Figure 3.** Trace plots for Model 1's posterior parameters.

In Figure 4, Gelman plots of Model 1's parameters are illustrated. The $y$-axis of the Gelman plot illustrates the shrink factor (or Gelman factor), which is used to estimate the relative between the variance between Markov chains and the variance within chains. Meanwhile, the $x$-axis demonstrates the iteration order of the simulation. As can be seen, the shrink factors of all parameters drop rapidly to 1 during the warm-up iterations (before the 2000th iteration), hinting that there is no divergence among Markov chains. Therefore, the Markov property is held.

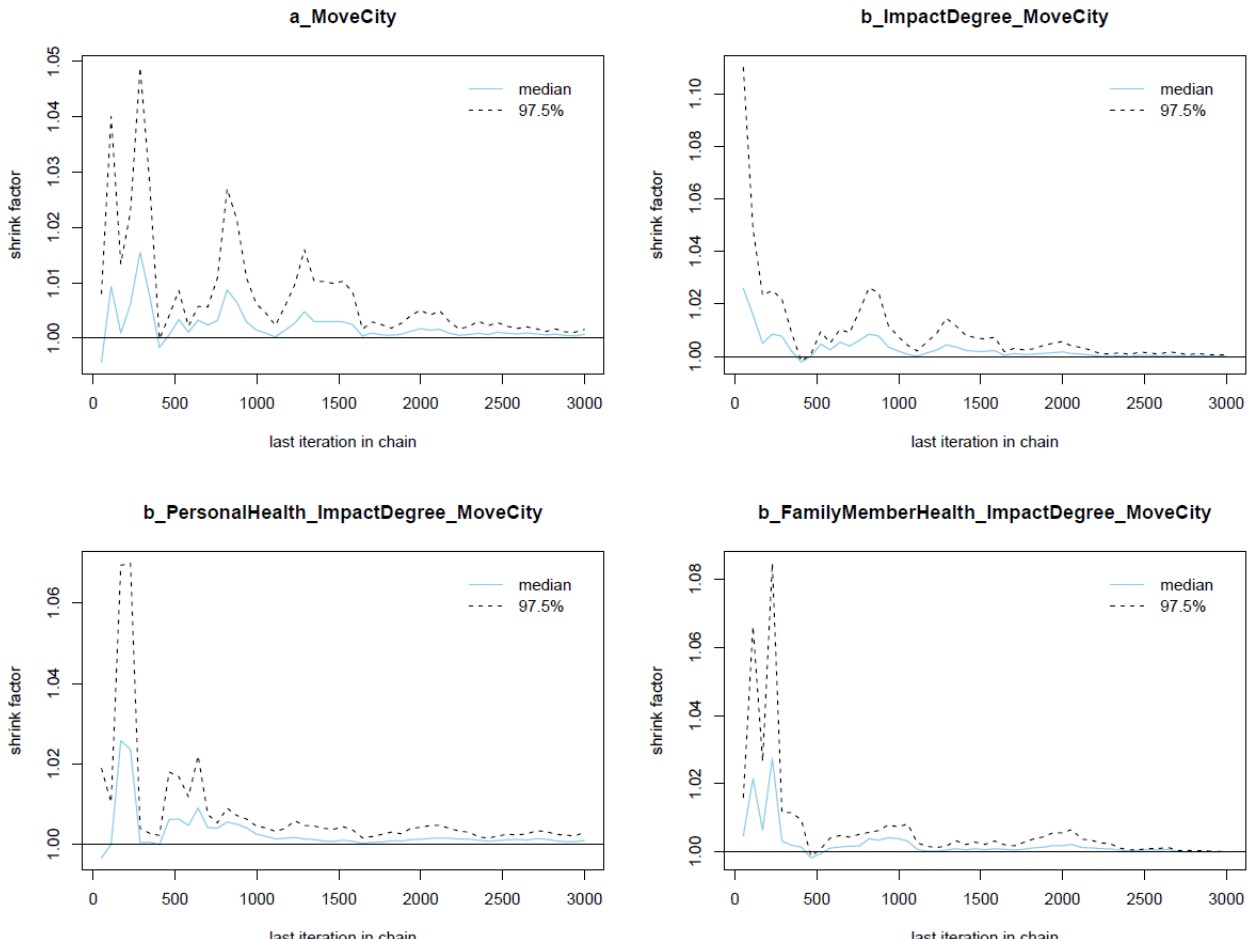

**Figure 4.** Gelman plots for Model 1's posterior parameters.

The next step in validating Model 1's convergence is visually diagnosing the Markov chains' autocorrelation levels (see Figure 5). The Markov chains' lag is illustrated on the $x$-axes of the plots, and the average level of autocorrelation of each chain is presented on the $y$-axes. Figure 5 shows that the average autocorrelation level drops rapidly after the fourth lag, inducing all parameters to acquire a large number of effective samples. The autocorrelation plots' illustration again confirms the convergence of Model 1's Markov chains.

From the simulated posterior results of Model 1, we found that the perceived impact of air pollution was negatively associated with the intention to immigrate to another city ($\mu_{ImpactDegree} = -0.48$ and $\sigma_{ImpactDegree} = 0.29$). This result confirms our assumption that the more an individual perceives the impacts of air pollution on their life, the more likely they will think of migrating to another city or country. However, the cost–benefit judgment of urban citizens about migration due to air pollution is much more complex. Perceived negative health effect caused by air pollution on family member increases the probability of domestic migration intention ($\mu_{FamilyMemberHealth*ImpactDegree} = 0.35$ and $\sigma_{FamilyMemberHealth*ImpactDegree} = 0.26$), whereas the effect of prior negative personal experiences attributed to air pollution is minimal and could be neglected ($\mu_{PersonalHealth*ImpactDegree} = 0.02$ and $\sigma_{PersonalHealth*ImpactDegree} = 0.23$).

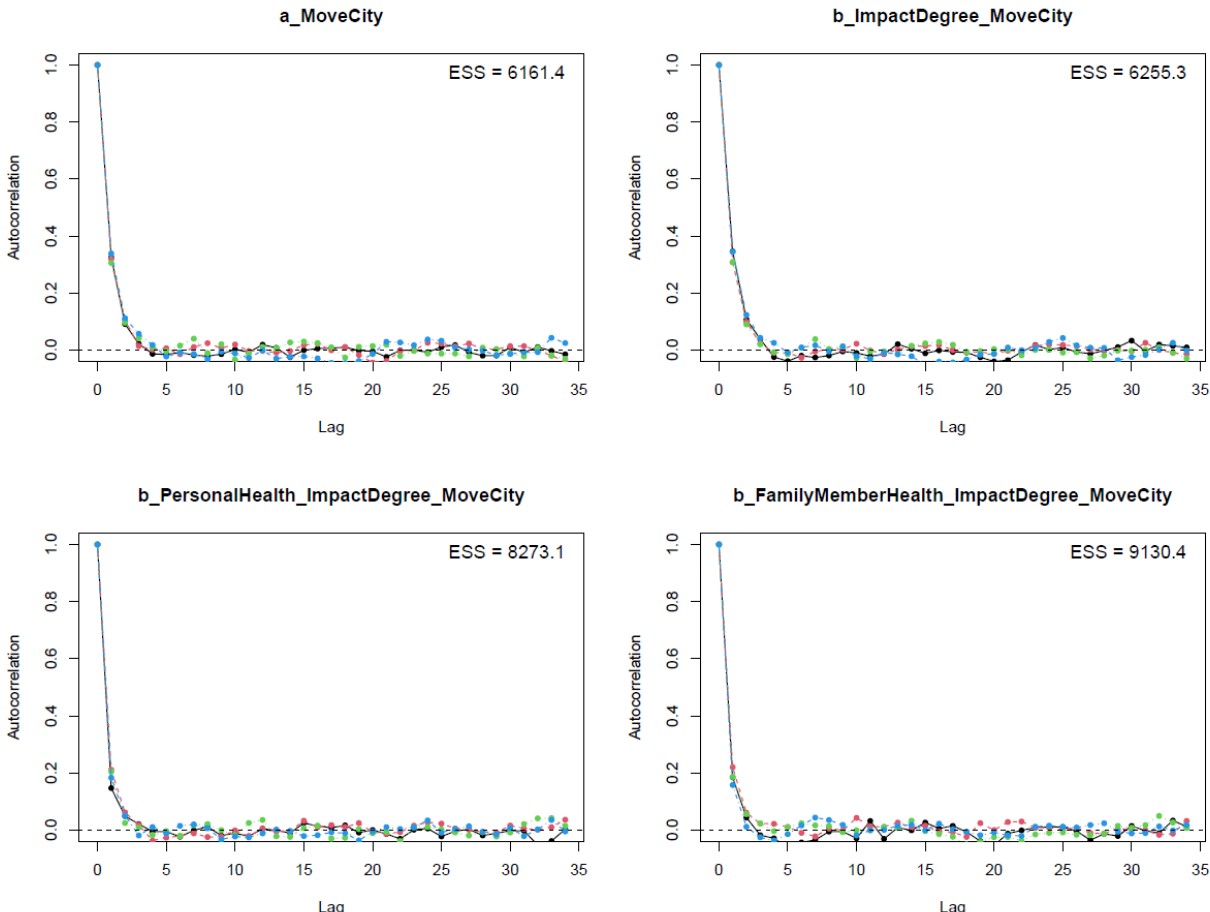

**Figure 5.** Autocorrelation plots for Model 1's posterior parameters.

For robustness check, prior-tweaking was performed. In both cases, demonstrating our beliefs and disbeliefs about the associations, the coefficients' effect patterns did not change, even though the degree slightly changed. Thus, it is conclusive that the effects in Model 1 are robust even when the prior beliefs vary.

The distributions of Model 1's parameters estimated with uninformative priors are visualized in the interval plot to assess their reliability (see Figure 6). The probability distributions of parameters are shown on the *x*-axis of the plot. Most of the distribution of *ImpactDegree* lies on the negative side of the axis, indicating a highly reliable negative association between *ImpactDegree* and *MoveCity*. The distribution of *FamilyMember-Health* ∗ *ImpactDegree* is located on the positive side, implying that a family member's perceived negative health effect had the highest probability of positively moderating the effect of perceived air pollution on domestic migration intention. It is shown in Figure 5 that the distribution of *PersonalHealth* ∗ *ImpactDegree* is near 0 and has a high standard deviation, so the moderation effect of *PersonalHealth* is not significant.

Based on the simulated posterior coefficients of Model 1 (see Table 2), we can use the logit model below to calculate the domestic migration intention probability. The parameters' mean values are used because they have the highest occurrence probability.

$$\ln \frac{\pi_{migration}}{\pi_{nomigration}} = -1.90 - 0.48 \times ImpactDegree + 0.02 \times PersonalHealth \times ImpactDegree$$
$$+ 0.35 \times FamilyMemberHealth \times ImpactDegree$$

(7)

Using this model, we can calculate as follows, for example, the domestic migration intention probability of a citizen perceiving strong impacts from air pollution, having experience of sickness both personally and that of a family member.

$$\pi_{migration} = \frac{e^{-1.90-0.48\times1+0.02\times1\times1+0.35\times1\times1}}{1+e^{-1.90-0.48\times1+0.02\times1\times1+0.35\times1\times1}} = 0.1182 = 11.82\% \tag{8}$$

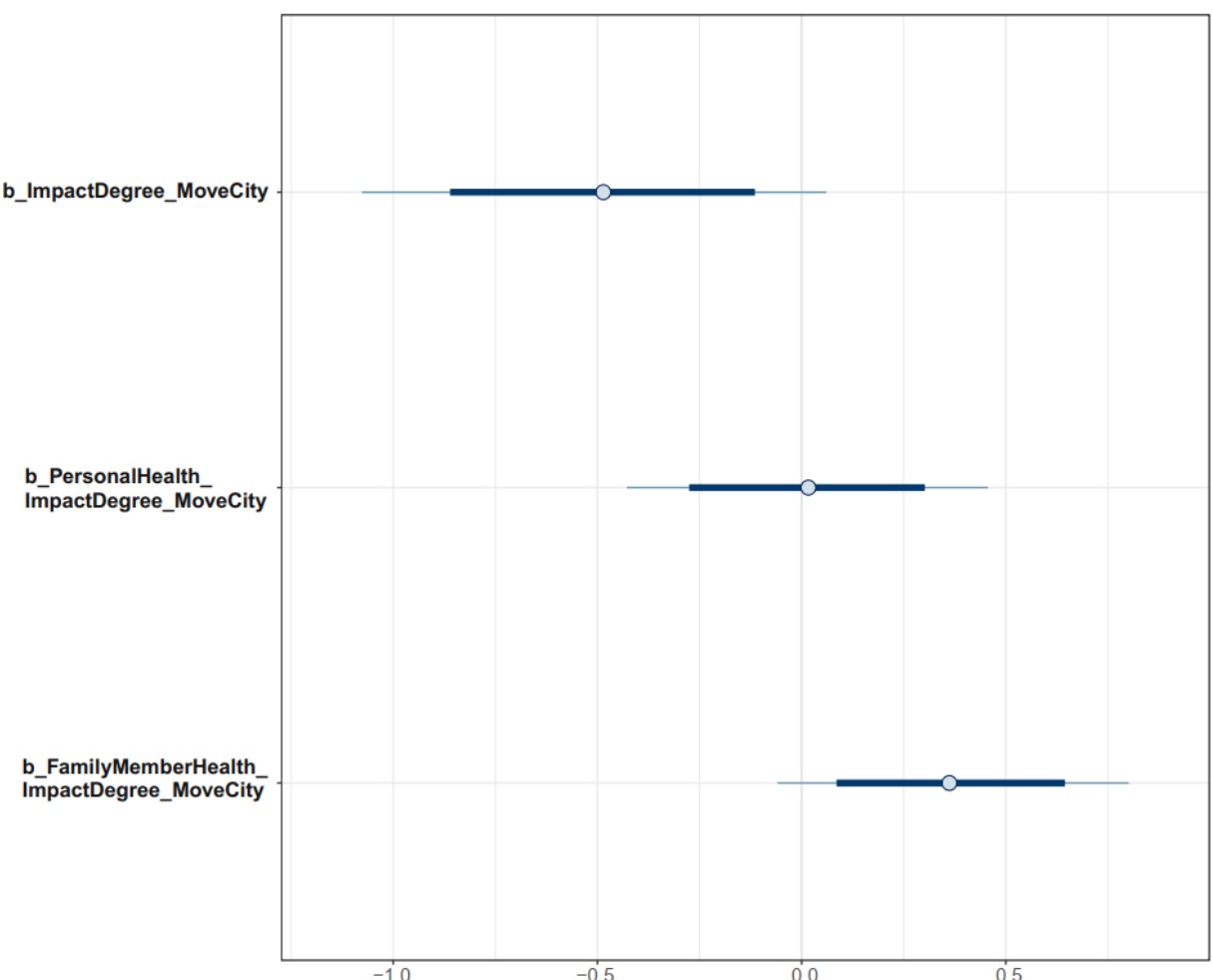

**Figure 6.** Distributions of Model 1's posterior coefficients estimated with uninformative priors.

The domestic migration intention probabilities across scenarios are visually demonstrated in Figure 7.

### 3.2. Model 2: Emigration Intention to Another Country with Lower Air Pollution Level

In the second model, we examined the perceived air pollution impact on the citizens' lives and its interactions with prior personal experience of respiratory disease. We perceived her/his family member's health effects caused by contaminated air on foreign migration intention. The logical network of Model 2 can be illustrated in Figure 8.

The model's goodness-of-fit is relatively high as no *k* values on the PSIS diagnostic plot are higher than 0.5 (see Figure 9).

The diagnostic statistics of Model 2 indicate that the posterior results are well convergent; the *n_eff* values are greater than 1000, and Rhat values are equal to 1 (see Table 3). The visual diagnostic methods, such as the trace plots (see Figure A1), the Gelman plots (see Figure A2), and the autocorrelation plots (see Figure A3), also confirm Model 2's Markov chains' convergence.

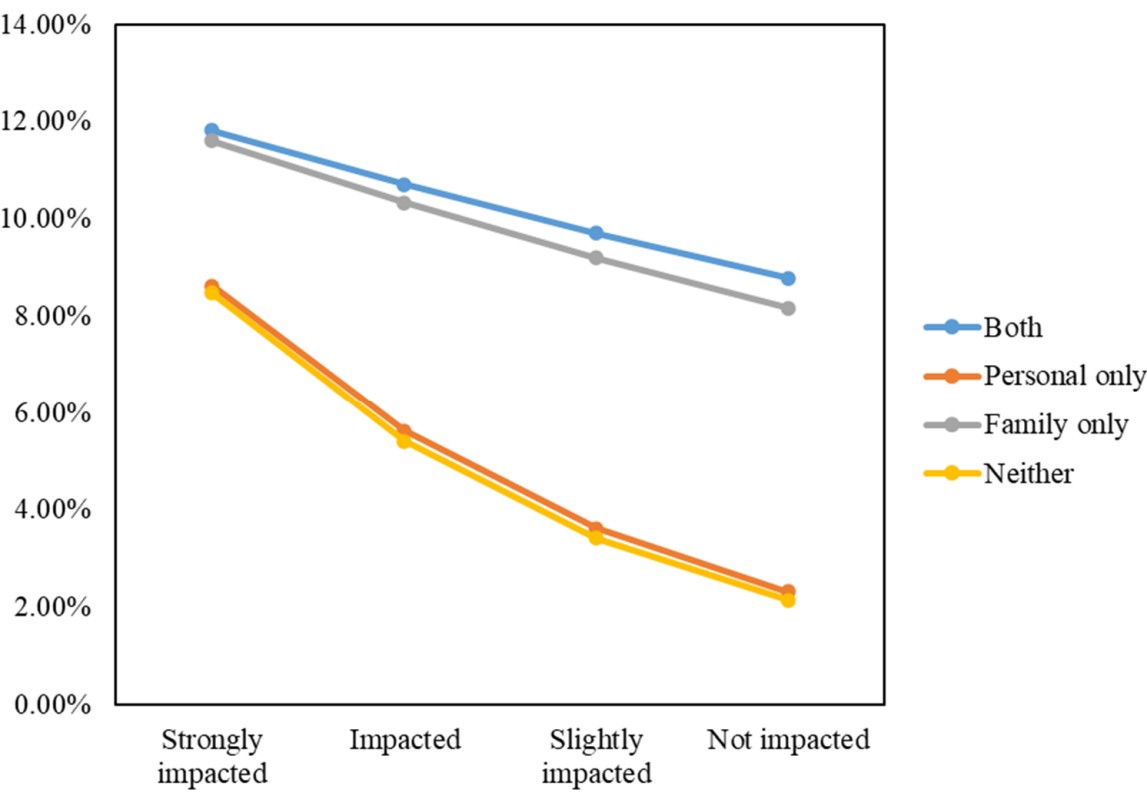

**Figure 7.** Domestic migration intention probabilities based on levels of perceived air pollution impact, personal sickness experience, and family member's sickness experience.

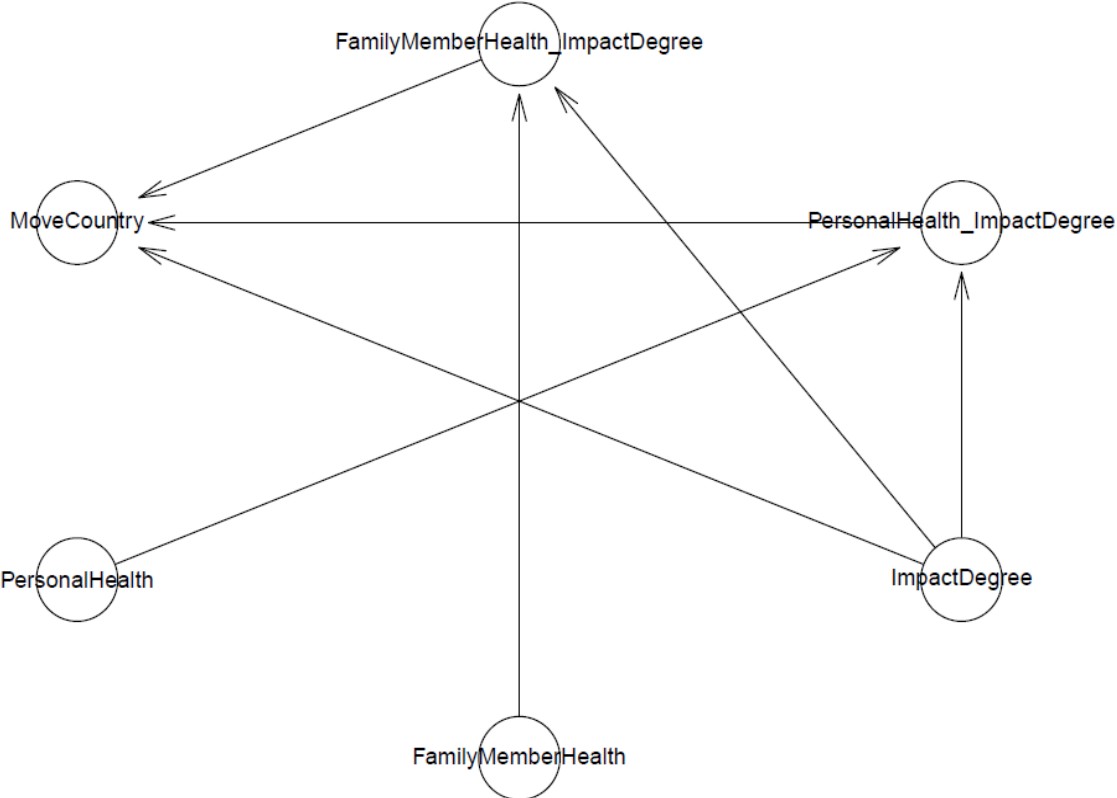

**Figure 8.** Model 2's logical network.

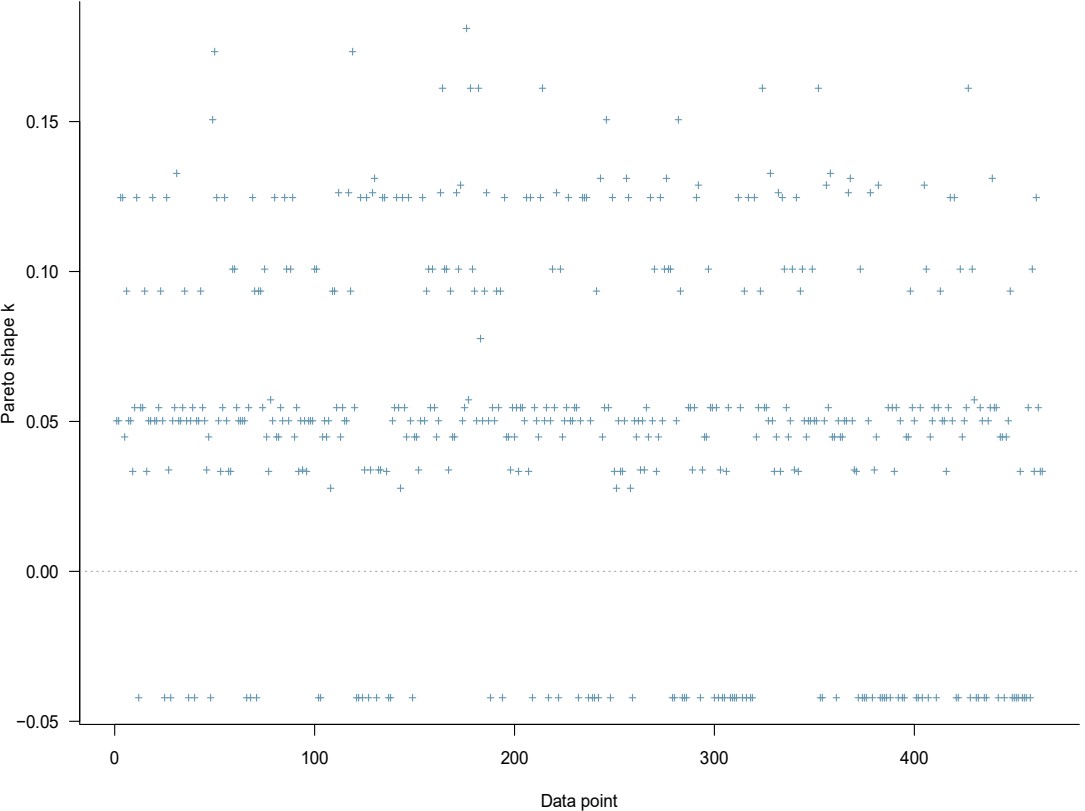

**Figure 9.** Model 2's PSIS diagnostic plot.

**Table 3.** Model 2's simulated posterior coefficients.

| Parameters | Uninformative | | Belief | | Disbelief | | n_eff | Rhat |
|---|---|---|---|---|---|---|---|---|
| | **Mean** | **SD** | **Mean** | **SD** | **Mean** | **SD** | | |
| *Constant* | −1.74 | 0.55 | −1.82 | 0.55 | −1.69 | 0.54 | 5261 | 1 |
| *ImpactDegree* | −0.78 | 0.36 | −0.86 | 0.37 | −0.76 | 0.34 | 5513 | 1 |
| *PersonalHealth* ∗ *ImpactDegree* | 0.42 | 0.28 | 0.48 | 0.24 | 0.31 | 0.24 | 7542 | 1 |
| *FamilyMemberHealth* ∗ *ImpactDegree* | −0.09 | 0.27 | 0.11 | 0.24 | −0.04 | 0.23 | 8015 | 1 |

Note: SD = Standard deviation; The effective sample size (*n_eff*) and Gelman value (*Rhat*) of simulated results with different priors are almost similar, so only the *n_eff* and *Rhat* of simulated results using uninformative priors are presented.

From the simulated posterior results of Model 2, we found that the perceived air pollution impact on life was negatively associated with the intention to migrate to a less polluted country ($\mu_{ImpactDegree} = -0.78$ and $\sigma_{ImpactDegree} = 0.36$). This result is similar to Model 1's outcome, confirming our assumption that the higher degree of perceived air pollution impacts, the higher the probability people intend to migrate abroad. However, the moderation effects of personal experience of health issues and her/his relative's health effects caused by air pollution are different from Model 1's outcomes. Prior negative personal experiences caused by air pollution increased the probability of foreign migration intention ($\mu_{PersonalHealth*ImpactDegree} = 0.42$ and $\sigma_{PersonalHealth*ImpactDegree} = 0.28$), whereas the moderation effect of *FamilyMemberHealth* on the relationship between *ImpactDegree* and *MoveCountry* was negligible ($\mu_{FamilyMemberHealth*ImpactDegree} = -0.09$ and $\sigma_{FamilyMemberHealth*ImpactDegree} = 0.27$). Using the prior-tweaking techniques, we also found no significant changes in the simulated posterior results of Model 2. All coefficients' probability distributions are presented in Figure 10.

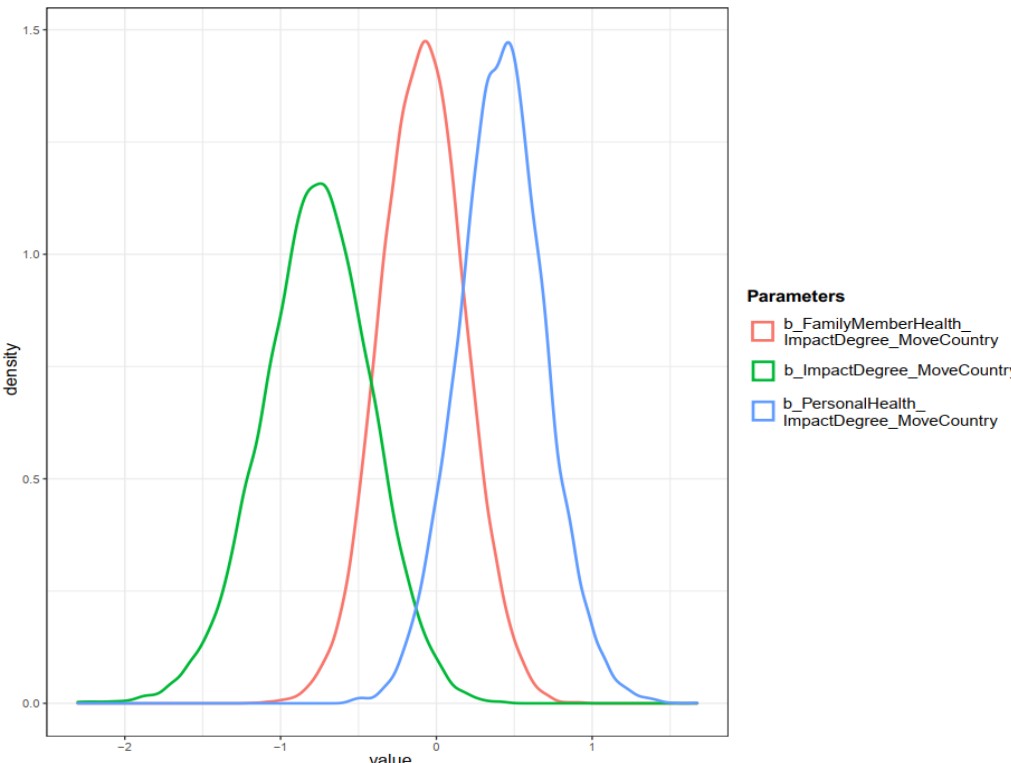

**Figure 10.** Distributions of Model 2's posterior coefficients on a density plot.

Using the same calculation method as with Model 1 presented above, we visually demonstrate international migration intention probabilities in Figure 11.

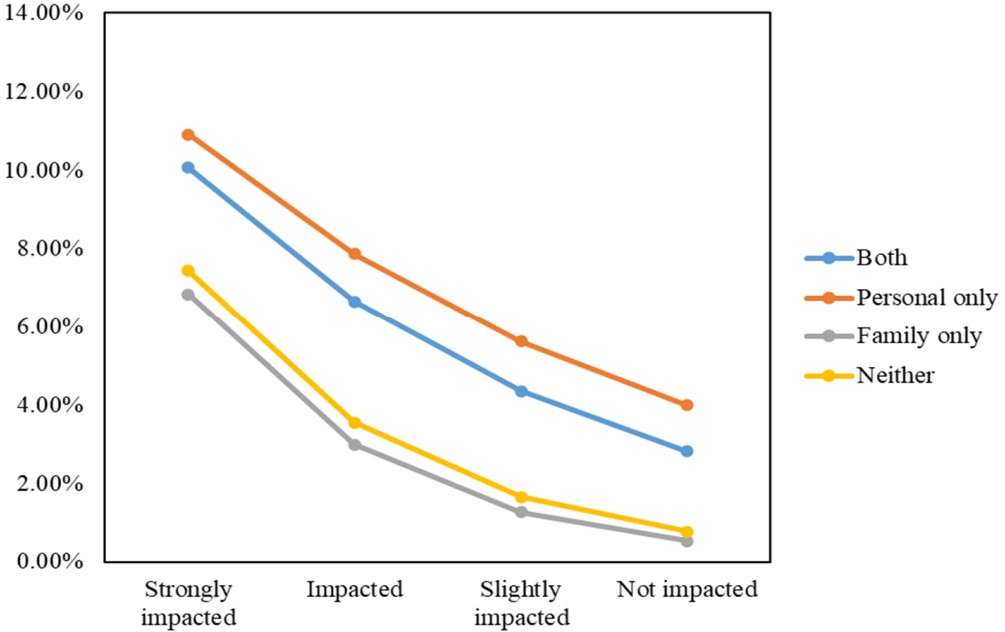

**Figure 11.** International migration intention probabilities based on levels of perceived air pollution impact, personal sickness experience, and family member's sickness experience.

## 4. Discussion

In this study, we employed the BMF to construct and analyze models to predict the domestic and foreign migration intention among Vietnamese urban residents. Estimated results based on 475 inhabitants in Hanoi—one of the most polluted capital cities in the

world—validate our assumptions that people perceiving more air pollution impacts would be more likely to migrate domestically and abroad. Our result is consistent with former studies on migration intention and behavior due to air quality [15,17,18,53,85–87]. Interestingly, we also found that although prior experience and acknowledgment of respiratory disease attributed to air pollution increase the migration probability, their effects on domestic and international migration intentions were inconsistent. Specifically, while prior experiences of respiratory disease caused by air pollution positively moderated the association between perceived air pollution impact and international migration intention, the acknowledgment of family members' sickness caused by air pollution did not. These moderation effects became the opposite when the intention was domestic migration.

One possible explanation for these strange results is that respondents' consideration of long-term income in our postulations was previously not sufficiently assessed. Migration decision is based not only on the quality of life—such as environmental factors—but also on economic opportunity perceptions [88]. In most cases, moving to another place and starting a new job can be considered a lifetime decision. It is not easy to make, regardless of whether the destination is domestic or international. In our earlier postulation in the Model Construction subsection, health was deemed a vital factor that could adversely affect inhabitants' well-being and short-term income (due to "discontinued operations"), which could eventually drive inhabitants to migrate. However, on the other side of the coin, the state of well-being could not be sustained in the long term if the economic conditions were not guaranteed. Thus, given that migration is a lifetime decision, migration ideation induced by air pollution would be more likely to happen when two main conditions are met: (1) the expected destination has a less polluted environment that could improve the individual's well-being; and (2) individual could acquire the economic opportunities in the expected destination to sustain their life and regain the cost spent on moving.

While the first condition can be easily met because many places within and beyond Vietnam are less polluted than Hanoi, the second condition seems more difficult to meet domestically since Hanoi's job opportunities are among the highest in Vietnam. In a prior study on Hanoi inhabitants' migration intention, we found that people's domestic migration probability is increased by the perceived availability of less polluted provinces; however, the effect shrinks if the distance cost is high [19]. Interpretively, health costs inflicted by air pollution might be a vital element that drives moving ideation, but economic factors cannot be separated from the cost–benefit judgment process of migration.

International migration is different. Markets in many high-income countries are more developed than those in Vietnam, so international migration can be perceived as a way to improve quality of life and acquire better job opportunities. A country's emigrant population generally rises until that country reaches the upper-middle-income level [89]. This might also explain why perceived air pollution impacts had a stronger effect on international migration intention than domestic migration intention, as shown in our results.

However, why could only the respondent's prior experience of respiratory disease improve the effect of perceived air pollution impacts on international migration intention, and their acknowledgment of a family member's sickness did not? The result somewhat reflects the traits of *Homo Oeconomicus*, who always act to maximize utilities given the perceived constraints and opportunities among respondents [41]. Assuming that domestic or international migration will result in better health outcomes, the migration decision will be greatly subject to economic factors [88]. Three major economic factors are involved in the cost–benefit judgment of the migration process: (1) job opportunity; (2) moving cost; and (3) income loss due to "discontinued operation" caused by air pollution. When a family member is sick because of air pollution, the individual's job is not discontinued, so it is plausible to say that the respondent is not sensitive to the income loss caused by "discontinued operation".

In contrast, when the individual experiences respiratory disease attributed to air pollution, their income is lost due to "discontinued operations". Their perception of income loss due to "discontinued operations" caused by air pollution is more transparent.

Therefore, income loss due to "discontinued operation" is more likely to be considered when the individual experiences air pollution-related disease than when knowing their family member is sick.

As presented above, the job opportunities in Hanoi are among the highest in Vietnam, so moving domestically might not generate an economic surplus. In this case, moving costs will be considered an exchange for better health. Suppose the individual personally experiences sickness and perceives income loss due to "discontinued operations". In this case, the domestic migration ideation will be less likely to emerge because they might perceive fewer opportunities to regain the lost income at the destination. When moving internationally, an expected economic surplus is much higher than that of domestic migration. The surplus often even bypasses the moving cost in circumstances of people migrating from low-income to high-income countries [89]. In addition, the expectation of no income loss due to "discontinued operations" due to air pollution and the opportunity to regain the previous loss in a foreign destination also adds more perceived benefit to the migration ideation during the cost–benefit judgment process. On the contrary, acknowledging family members' sickness induced by air pollution does not influence the international migration intention because the individual feels no pressure about the income loss due to "discontinued operation".

The above explanations only focus on the economic aspect of the cost–benefit judgment to highlight some traits of *Homo Oeconomicus* among the respondents, so we assume domestic and international migration generate the same level of health-related benefit. Nevertheless, foreign countries that are economically more developed also have better living standards, including environmental quality, such as some North American and European countries, Japan, South Korea, etc. Comparatively, Vietnam is only ranked 141th on Environmental Performance Index [90]. As a result, besides economic factors, the ideal living conditions in developed countries compared to domestic cities also substantially contribute to the migration decision due to air pollution.

Based on the presented results and discussion, we advocate that environmental issues should not only be examined within the boundary of strictly environmental science discipline, but the consideration needs to encompass economics and sociology. Indeed, career opportunities and labor capacity considerations concerning pollution-induced migration tendencies are also reflected in the differences among age groups, genders, and educational levels [15,53,91]. Such integral considerations are aligned with Lee's migration theory emphasizing individual situations and perceptions [24]. Certain similarities in terms of harm-avoidance behavior can be seen in the case of conflict-induced migration, where migrating action is driven by both motivation and perceived opportunity [92].

The cost–benefit judgments of socioeconomic values may appear seemingly trivial on a collective level, but they can be facilitated by environmental stressors as catalysts, which might eventually lead to mass migration tendencies, especially when low- and middle-income countries are more severely affected by not only air pollution but also climate change problems [93]. To prevent the environmental problems from being exacerbated, besides political interventions and technological infrastructure development, it is necessary to replace the current "eco-deficit culture" with the "eco-surplus culture", especially in the private sector [94].

Our study's limitations are presented as follows. First of all, the explanation of the estimated results using the traits of *Homo Oeconomicus* is speculative, so further studies on migration due to environmental stressors are needed to examine the associations of *Homo Oeconomicus* characteristics with migration decisions. Secondly, the samples were collected only in an urban area, so the study findings might not be representative of the associations of health and economic factors with migration intention among rural residents. Moreover, as our interpretations were based on Hanoi's environmental and economic development levels, the impacts of health and economic factors on the migration consideration should be validated using samples from other urban areas in developing countries, not limited to Vietnam. The answers of respondents were largely influenced by the environmental conditions that they were experiencing. Given that the concentration of air pollutants

in Hanoi has seasonal variation [95], future studies should be conducted to replicate the research in various seasons for validation.

## 5. Conclusions

We provide empirical evidence from data of Hanoi citizens that risk perception of air pollution increases the probability of migration intention, which is consistent with other former studies on pollution-induced migration. However, our results also show that there are differences in the health risk perception information types (information source being personal experience or family member's experience) for predicting domestic and international migration intention. A deeper analysis reveals that such differences are due to the subjective cost–benefit evaluation in each scenario regarding the perceived value of the act of migrating. This information process has *Homo Oeconomicus* traits, expressed as the personal net benefit optimization in an individual's thinking. Our study also shows that using the mind sponge information processing approach, seemingly inconsistent patterns in new findings can be effectively explained in alignment with existing theories.

**Author Contributions:** Conceptualization: Q.-H.V.; Methodology: M.-H.N., Q.-H.V. and T.-T.L.; Formal analysis and investigation: Q.-H.V., T.-T.L. and V.-P.L.; Writing—original draft preparation: M.-H.N., T.-T.L. and T.-T.V.; Writing—review and editing: M.-H.N., T.-T.L., V.-P.L. and T.-T.V.; Validation: Q.-H.V.; Resources: V.-P.L.; Supervision: Q.-H.V. All authors have read and agreed to the published version of the manuscript.

**Funding:** This research received no external funding.

**Data Availability Statement:** The data that support the findings of this study are available on The Open Science Framework for later replications (https://osf.io/us5tr/ (accessed on 3 July 2022)).

**Conflicts of Interest:** The authors declare no conflict of interest.

## Appendix A

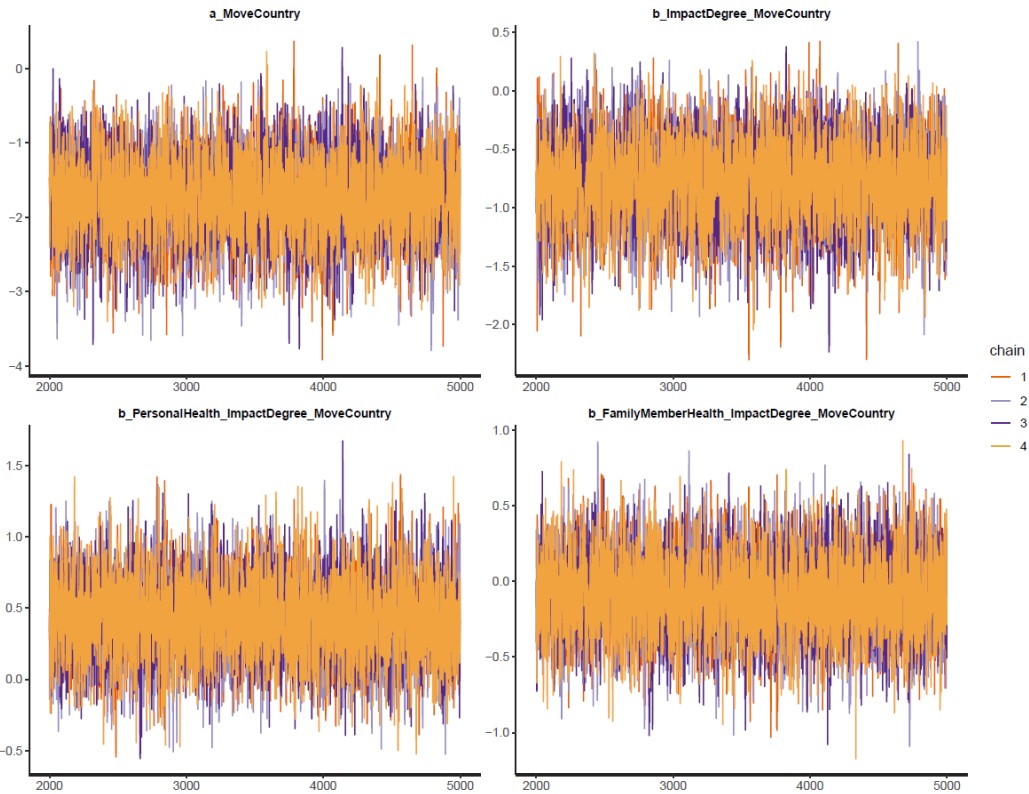

**Figure A1.** Trace plots for Model 2's posterior parameters.

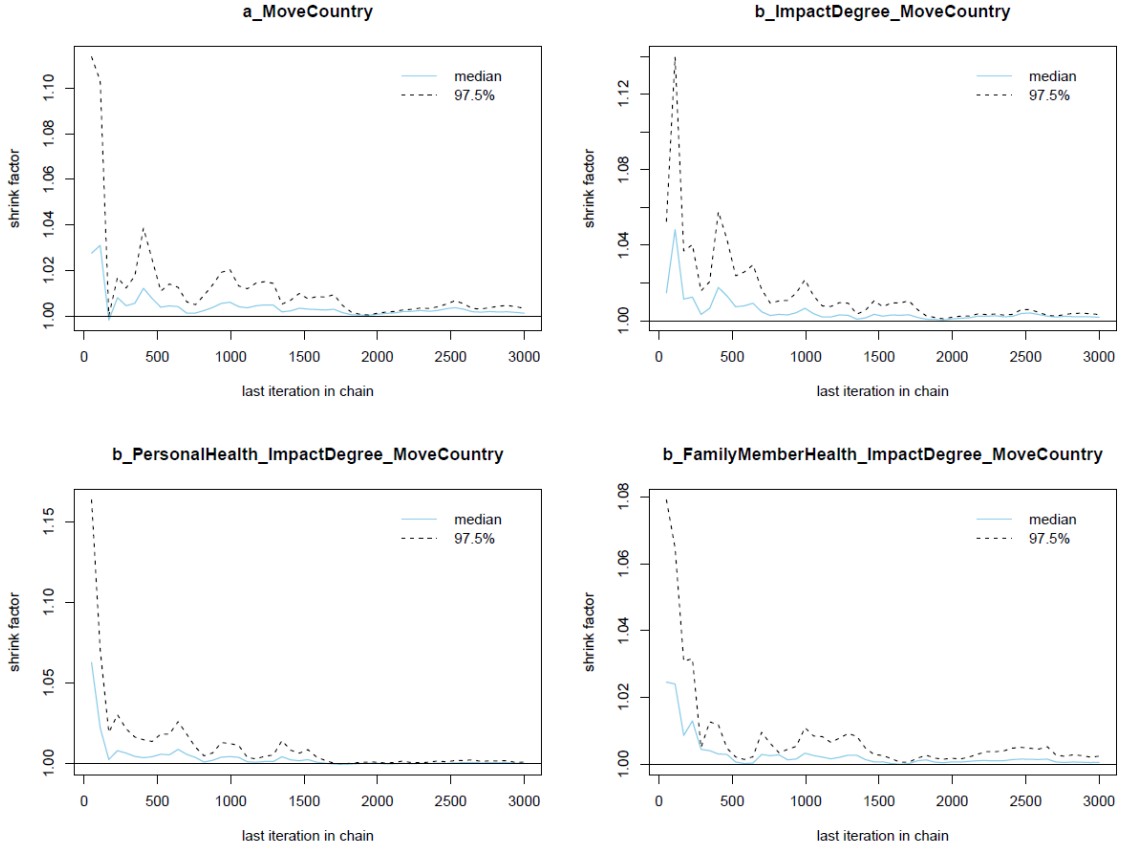

**Figure A2.** Gelman plots for Model 2's posterior parameters.

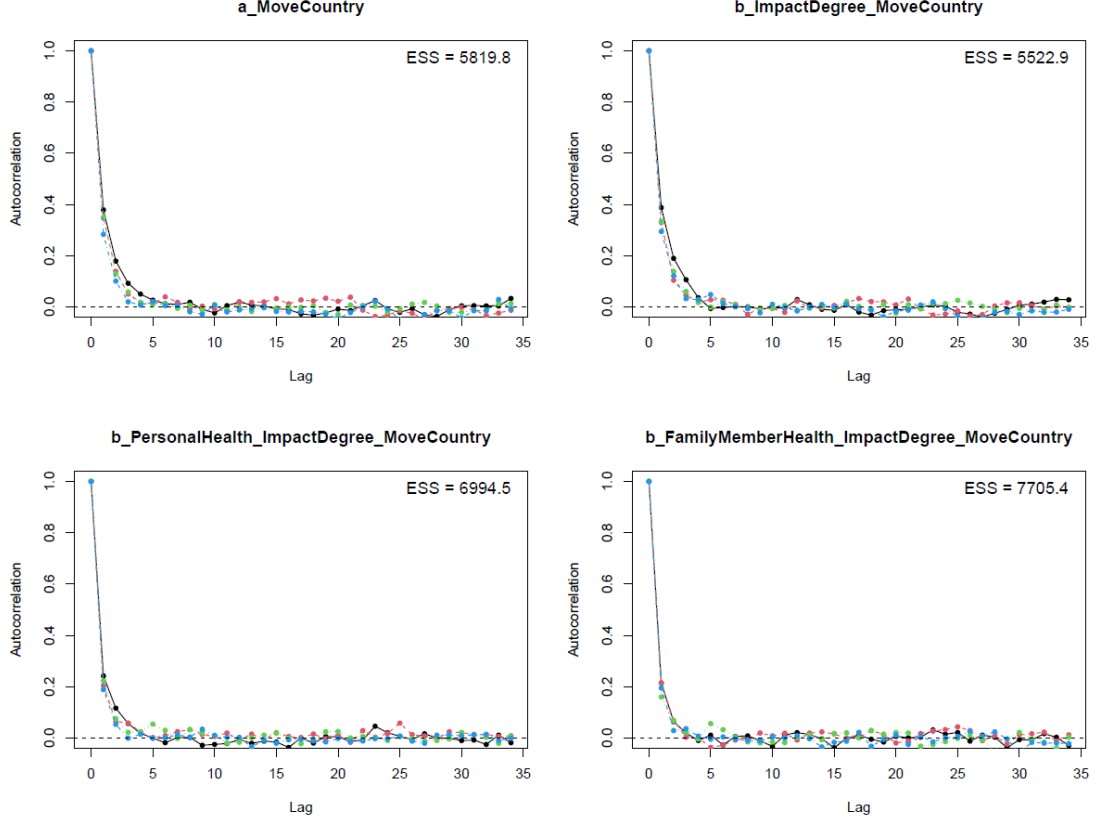

**Figure A3.** Autocorrelation plots for Model 2's posterior parameters.

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
