# Peer review of "Investigation into the Rationale of Migration Intention Due to Air Pollution Integrating the Homo Oeconomicus Traits"

_urbansci, doi:10.3390/urbansci7020059_

Round 1

Reviewer 1 Report

The rationale of migration intention combined with air pollution was fully investigated based on the Bayesian Mindsponge Framework among 475 inhabitants in Hanoi, Vietnam. This paper aims to deepen the understanding of the ideation of migrating behavior in terms of human information processing within socio-economic contexts. The topic is of interest to the community and has its significance. The structure and discussions in this article are clear and concise, but the figures and formats need revisions to better show the results. I recommend the acceptance for publication after minor revisions. Several editorial comments for improving the information content and presentation of the paper are listed as follows:

1. Keywords: The meanings expressed by environmental stress and air pollution are duplicated, so consider replacing environmental with BMF.

2. Introduction: The research on the specific hazards of air pollution to humans needs to be added.

3. Line 246 and 327: Tables in the manuscript should use a three-line table.

4. Line 294: PSIS-LOO is missing full name, please add.

5. As seasonal variations can affect the concentration of air pollution, will population migration have some seasonal variation?

6. Line 427: The legend in the figure 10 is not properly expressed, why is a colored box used to represent the curve?

7. Line 477-487: The conclusion lacks literature support, please add references.

8. Line 586-587: Please revise the reference format according to the journal's requirements.

Minor editing of English language required

Author Response

Thank you very much for your constructive comments! We have revised our manuscript according to your suggestions. Please see the attached file for detailed responses

Reviewer 2 Report

Dear authors,

thank you for this very well done article, which I read with great interest. The results of your study have a very high value for research and show a high topicality and relevance against the background of the growing air pollution problems in the world. Although I am not well familiar with the method you used (BMF), I was able to follow your calculations and methodological explanations well. I did not notice any gross inadequacies or errors. However, I was somewhat irritated that you did not use control variables in your analysis. Is this not feasible with the data or does the BMF method not make it possible? If the methodological approach allows it, I would highly recommend the inclusion of control variables. Also because I believe that the decision to migrate depends on many additional factors that were not captured in the analysis so far. I have also seen that your explanatory approach through homo economicus is not supported by the data in places and is therefore speculative. But you have already mentioned satisfactorily in the limitations that your reasoning is speculative in some places. So I have no problem with that. In terms of language and form, I did not notice any serious flaws. However, I still have one point of minor importance. I am not convinced by the argumentation in lines 302 to 306. Since the passage does not add much value to the rest of the paper anyway, I would advise removing it.

Overall, I think your contribution is well made.

Author Response

Thank you very much for your constructive comments! Please see the attached file for detailed responses

Reviewer 3 Report

This study investigates the effects of air pollution on the domestic and international migration intentions among 475 inhabitants in Hanoi, Vietnam. I have two major comments. I hope these comments can help the authors improve the paper.

1. The contribution of this study is unclear. The research topic is not new, because some previous studies have also analyzed the effects of air pollution on migration intentions. In the paper, several paragraphs should be added to provide a clear literature review on those relevant studies. The literature review should explicitly summarize the findings of previous studies and the relationship between this paper and previous studies. In addition, some sentences should be added to tell the readers what new findings are provided in this paper and the academic contribution of this paper.

2. Equations (1.1) and (2.1) should be modified. In these two equations, it is stated that the variables “MoveCity” and “MoveCountry” follow a normal distribution. However, as these two variables are 0-1 binary variables, they should not follow a normal distribution.

I think the language quality is generally fine.

Author Response

(The authors gave the same response as above.)

Round 2

Reviewer 3 Report

The authors have made some revisions. Regarding my previous comment on Equations (1.1) and (2.1), I still have some words to say. In these two equations, the variables are “MoveCity” and “MoveCountry”. By definition, they can only take the value 0 or 1. No matter how you sample, they can only be either 0 or 1. Thus, they cannot follow a normal distribution.

In fact, in Equation (3), it is written that the dependent variable in the regression equation is relevant to the occurrence probability (ln[p/(1-p)]), not the mean of a normal distribution.

Author Response

Thank you very much for your comment! We have provided a more thorough explanation for your concern regarding the distribution. Please see our detailed response in the attached file
